# G protein-regulated endocytic trafficking of adenylyl cyclase type 9

**André M Lazar[1], Roshanak Irannejad[2], Tanya A Baldwin[3], Aparna B Sundaram[4], J Silvio Gutkind[5], Asuka Inoue[6], Carmen W Dessauer[3], Mark Von Zastrow[2,7]\***

[1]Program in Biochemistry and Cell Biology, University of California San Francisco, San Francisco, United States; [2]Cardiovascular Research Institute and Department of Biochemistry and Biophysics, University of California San Francisco, San Francisco, United States; [3]Department of Integrative Biology and Pharmacology, University of Texas Health Science Center, Houston, United States; [4]Lung Biology Center, Department of Medicine, University of California San Francisco, San Francisco, United States; [5]Department of Pharmacology and Moores Cancer Center, University of California San Diego, San Diego, United States; [6]Graduate School of Pharmaceutical Sciences, Tohoku University, Aoba-ku, Sendai, Japan; [7]Department of Psychiatry and Department of Cellular and Molecular Pharmacology, University of California San Francisco, San Francisco, United States

**Abstract** GPCRs are increasingly recognized to initiate signaling via heterotrimeric G proteins as they move through the endocytic network, but little is known about how relevant G protein effectors are localized. Here we report selective trafficking of adenylyl cyclase type 9 (AC9) from the plasma membrane to endosomes while adenylyl cyclase type 1 (AC1) remains in the plasma membrane, and stimulation of AC9 trafficking by ligand-induced activation of Gs-coupled GPCRs. AC9 transits a similar, dynamin-dependent early endocytic pathway as ligand-activated GPCRs. However, unlike GPCR traffic control which requires β-arrestin but not Gs, AC9 traffic control requires Gs but not β-arrestin. We also show that AC9, but not AC1, mediates cAMP production stimulated by endogenous receptor activation in endosomes. These results reveal dynamic and isoform-specific trafficking of adenylyl cyclase in the endocytic network, and a discrete role of a heterotrimeric G protein in regulating the subcellular distribution of a relevant effector.

**\*For correspondence:**
mark@vzlab.org

**Competing interests:** The authors declare that no competing interests exist.

## Introduction

G protein-coupled receptors (GPCRs) comprise nature's largest family of signaling receptors and an important class of therapeutic targets (*Lefkowitz, 2007*; *Rosenbaum et al., 2009*). GPCRs are so-named because a major mechanism by which they mediate transmembrane signaling is through ligand-dependent activation of heterotrimeric G proteins that act as intracellular signal transducers (*Gilman, 1987*; *Hilger et al., 2018*; *Spiegel, 1987*; *Stryer and Bourne, 1986*; *Sunahara et al., 1996*). This conserved signaling cascade invariably requires one additional component, an 'effector' protein which is regulated by the G protein to convey the signal downstream (*Dessauer et al., 1996*; *Gilman, 1987*; *Rosenbaum et al., 2009*). Ligand-activated GPCR - G protein - effector cascades were thought for many years to be restricted to the plasma membrane, with endocytosis considered only in the context of signal termination and homeostatic down-regulation of receptors. This view has expanded over the past several years, driven by accumulating evidence that ligand-dependent GPCR and G protein activation processes can also occur from internal membrane compartments including endosomes (*Di Fiore and von Zastrow, 2014*; *Irannejad et al., 2015*; *Jong et al.,*

**eLife digest** Cells sense changes in their chemical environment using proteins called receptors. These proteins often sit on the cell surface, detecting molecules outside the cell and relaying messages across the membrane to the cell interior. The largest family of receptors is formed of 'G protein-coupled receptors' (or GPCRs for short), so named because they relay messages through so-called G proteins, which then send information into the cell by interacting with other proteins called effectors. Next, the receptors leave the cell surface, travelling into the cell in compartments called endosomes. Researchers used to think that this switched the receptors off, stopping the signaling process, but it is now clear that this is not the case. Some receptors continue to signal from inside the cell, though the details of how this works are unclear.

For signals to pass from a GPCR to a G protein to an effector, all three proteins need to be in the same place. This is certainly happening at the cell surface, but whether all three types of proteins come together inside endosomes is less clear. One way to find out is to look closely at the location of effector proteins when GPCRs are receiving signals. One well-studied effector of GPCR signaling is called adenylyl cyclase, a protein that makes a signal molecule called cAMP. Some G proteins switch adenylyl cyclase on, increasing cAMP production, while others switch it off.

To find out how GPCRs send signals from inside endosomes, Lazar et al tracked adenylyl cyclase proteins inside human cells. This revealed that a type of adenylyl cyclase, known as adenylyl cyclase 9, follows receptors as they travel into the cell. Under the influence of active G proteins, activated adenylyl cyclase 9 left the cell surface and entered the endosomes. Once inside the cell, adenylyl cyclase 9 generated the signal molecule cAMP, allowing the receptors to send messages from inside the cell. Other types of adenylyl cyclase behaved differently. Adenylyl cyclase 1, for example, remained on the cell surface even after its receptors had left, and did not signal from inside the cell at all.

Which cell behaviors are triggered from the membrane, and which are triggered from inside the cell is an important question in drug design. Understanding where effector proteins are active is a step towards finding the answers. This could help research into diseases of the heart, the liver and the lungs, all of which use adenylyl cyclase 9 to send signals.

*2018*; *Lobingier and von Zastrow, 2019*; *Lohse and Calebiro, 2013*; *Lohse and Hofmann, 2015*; *Vilardaga et al., 2014*).

The beta-2 adrenergic receptor (β2AR) provides a clear example, and is generally considered a model for the GPCR family more broadly (*Lefkowitz, 2007*; *Rosenbaum et al., 2009*). β2ARs initiate signaling in response to binding of an agonist ligand by activating ('coupling' to) the stimulatory heterotrimeric G protein, Gs, at the plasma membrane. β2ARs then internalize through agonist-dependent accumulation into clathrin-coated pits, efficiently recycle and transit continuously between the plasma membrane and endosomes in the prolonged presence of agonist (*von Zastrow and Kobilka, 1994*; *von Zastrow and Kobilka, 1992*). Agonist-induced clustering of β2ARs into coated pits, the process initiating this cycle, is promoted by receptor phosphorylation and binding to β-arrestins at the plasma membrane (*Ferguson et al., 1996*; *Goodman et al., 1996*). These events were shown previously to inhibit β2AR coupling to Gs (*Lohse et al., 1990*), and β2AR inactivation was recognized to precede removal from the cell surface even before this mechanistic elucidation (*Harden et al., 1980*). Accordingly, it was believed for many years that β2ARs are unable to engage G proteins once internalized. This view changed with the finding that β2ARs reacquire functional activity shortly after arriving in the limiting membrane of early endosomes, and then activate Gs again from this location (*Irannejad et al., 2013*). A number of GPCRs have now been shown to activate Gs after endocytosis, initiating a discrete 'second wave' of downstream signaling which varies in magnitude and duration depending (in part) on the residence time of activated receptors in the endocytic network (*Irannejad et al., 2015*; *Lohse and Calebiro, 2013*; *Thomsen et al., 2016*; *Tian et al., 2016*; *Varandas et al., 2016*; *Vilardaga et al., 2014*).

Gs transduces downstream signaling by stimulating adenylyl cyclases (ACs) to produce cyclic AMP (cAMP), an important diffusible mediator (*Lohse and Hofmann, 2015*; *Sutherland, 1971*; *Taylor et al., 2013*), and studies of cAMP signaling provided much of the initial evidence supporting

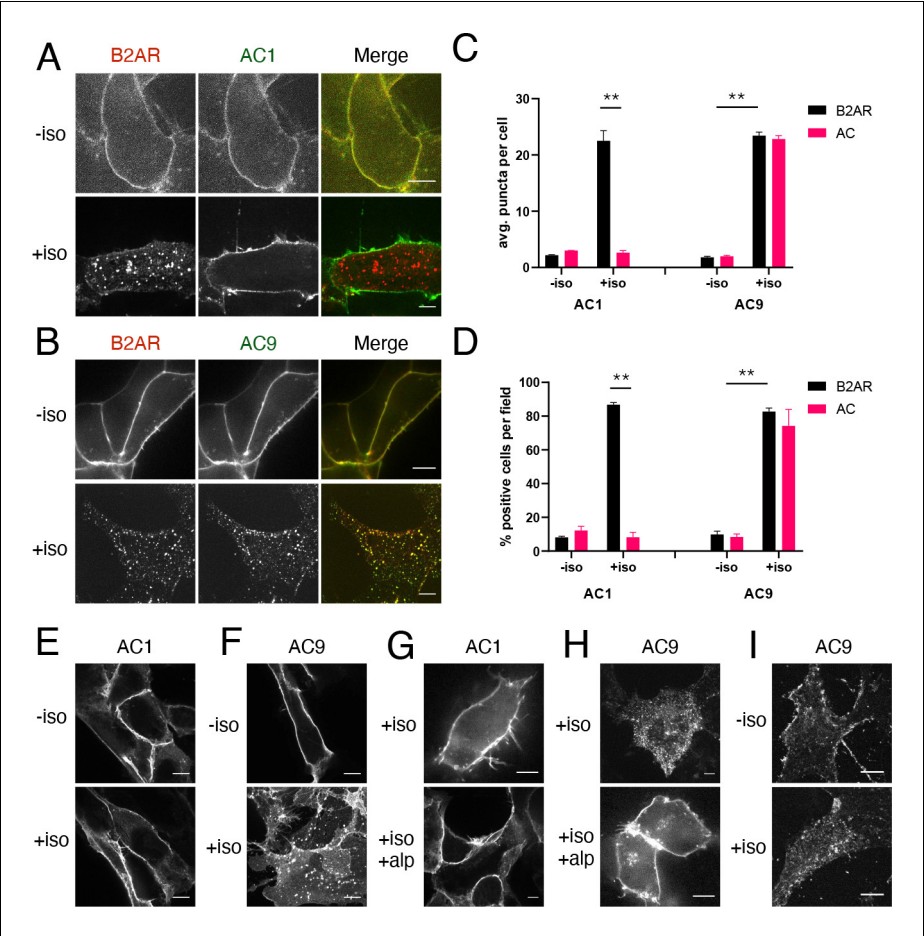

**Figure 1.** β2AR activation promotes redistribution of AC9 but not AC1. (**A**) Representative confocal imaging of HEK293 cells coexpressing HA-β2AR and Flag-AC1 after treatment with 10 μM isoproterenol or control for 30 min. Scale Bar is 8 μm. (**B**) Representative confocal images of HEK293 cells coexpressing HA-β2AR and Flag-AC9 after treatment with 10 μM isoproterenol or control for 30 min. Scale Bar is 8 μm. (**C**) Quantification of internal puncta that are β2AR or AC1/9 positive, taken from wide field images (see *Figure 1—figure supplement 1D and E*) [mean ± SEM; n = 3 experiments, 10 visual fields and 200+ cells per condition]. **p<0.01 by two-tailed t-test. (**D**) Quantification of cells with >10 internal puncta that are β2AR or AC1/9 positive, taken from wide field images (see *Figure 1—figure supplement 1D andE*) [mean ± SEM; n = 3 experiments, 10 visual fields and 200+ cells per condition]. **p<0.01 by two-tailed t-test. (**E–F**) Representative confocal imaging of HEK293 cells expressing Flag-AC1 (**E**) or Flag-AC9 (**F**) after treatment with 10 μM isoproterenol or control for 30 min. Scale Bar is 8 μM. (**G–H**) Representative confocal imaging of HEK293 cells expressing Flag-AC1 (**G**) or Flag-AC9 (**H**). Cells were stimulated with 100 nM isoproterenol for 30 min with or without 15 min of pretreatment with 10 μM alprenolol. (**I**) Representative confocal images of primary culture human airway smooth muscle cells immunostained for endogenous AC9 after treatment with 10 μM isoproterenol or control for 30 min. Scale Bar is 16 μm.

The online version of this article includes the following figure supplement(s) for figure 1:

**Figure supplement 1.** Functional validation of Flag-tagged AC9 and quantification of isoform-specific localization to endosomes.

the potential of GPCRs to mediate ligand-dependent signaling via G proteins after endocytosis (*Calebiro et al., 2009*; *Clark et al., 1985*; *Ferrandon et al., 2009*; *Kotowski et al., 2011*; *Mullershausen et al., 2009*; *Slessareva et al., 2006*). Nine transmembrane AC isoforms are conserved in mammals, each stimulated by Gs but differing in regulation by other G proteins and signaling intermediates, and multiple AC isoforms are typically coexpressed in tissues (*Sadana and Dessauer, 2009*; *Sunahara et al., 1996*; *Wang et al., 2019*). Biochemical and structural aspects of

regulated cAMP production by ACs have been extensively studied but much less is known about the cellular biology of ACs.

According to the present understanding, GPCR-stimulated cAMP production requires all three 'core' components of the signaling cascade – the GPCR, Gs and AC – to be in the same membrane bilayer (*Gilman, 1989*). Whereas GPCRs and Gs are well known to undergo dynamic redistribution between the plasma membrane and endocytic membranes (*Allen et al., 2005*; *Hynes et al., 2004*; *Irannejad et al., 2015*; *Marrari et al., 2007*; *von Zastrow and Kobilka, 1992*; *Wedegaertner et al., 1996*), the subcellular localization and trafficking properties of transmembrane ACs remain largely unknown. Nevertheless, adenylyl cyclase activity was noted on intracellular membranes many years ago (*Cheng and Farquhar, 1976*) and more recent studies have implicated several AC isoforms in GPCR-regulated cAMP production from endomembrane compartments (*Calebiro et al., 2009*; *Cancino et al., 2014*; *Ferrandon et al., 2009*; *Inda et al., 2016*; *Kotowski et al., 2011*; *Vilardaga et al., 2014*). Key knowledge gaps at the present frontier are how ACs localize to relevant internal membranes, if the subcellular localization of ACs is selective between isoforms, and if this localization is regulated. Here we report initial inroads into this frontier by demonstrating the dynamic and isoform-specific endocytic trafficking of AC9, and its ligand-dependent regulation through Gs.

## Results

### Regulated and isoform-selective trafficking of AC9 to endosomes

Human embryonic kidney (HEK293) cells comprise a well established model system for investigating GPCR signaling via the cAMP cascade, particularly signaling initiated by β2ARs which are endogenously expressed in these cells (*Violin et al., 2008*). The β2AR stimulates cAMP production primarily from the plasma membrane, with endosomal activation contributing a relatively small but functionally significant fraction (*Irannejad et al., 2013*; *Tsvetanova and von Zastrow, 2014*). In a similar vein, AC3 and AC6 are the most highly expressed AC isoforms and are major producers of global cAMP in these cells, while AC1 and AC9 transcripts are expressed at only moderately reduced levels and comparable to one another (*Soto-Velasquez et al., 2018*). This prompted us to ask whether AC1 and/or AC9 might be relevant to generating the fraction of cellular cAMP produced by β2AR activation in endosomes.

As a first step to investigate this hypothesis, we examined the subcellular localization of AC1 and AC9 using a recombinant epitope tagging strategy, beginning with AC1 because this isoform was shown previously to tolerate an N-terminal Flag tag (*Chen et al., 1997*). When expressed in HEK293 cells, Flag-tagged AC1 (Flag-AC1) localized primarily to the plasma membrane, similar to a co-expressed HA-tagged β2AR construct (HA-β2AR). Application of the β-adrenergic agonist isoproterenol promoted HA-β2ARs to redistribute within minutes to cytoplasmic punctae, as described previously, but Flag-AC1 remained at the plasma membrane (*Figure 1A*). While clearly resolved in confocal sections, this difference in localization was sufficiently strong to be evident in lower-magnification widefield images surveying many cotransfected cells (*Figure 1—figure supplement 1D*). We assessed reproducibility of this effect in two ways. First, we determined the number of internal punctae per cell in which HA-β2AR and Flag-AC1 colocalized. Second, we determined the fraction of cells visualized in

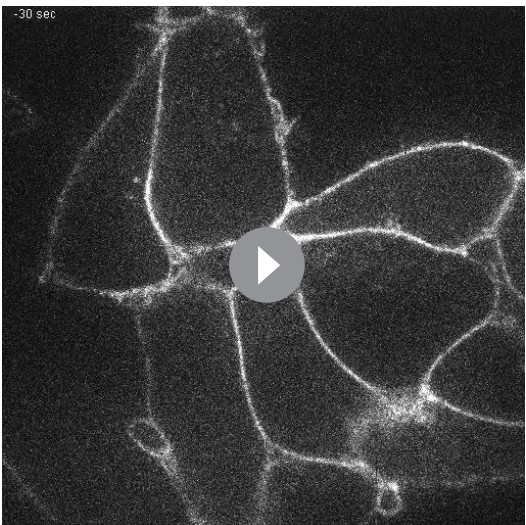

**Video 1.** This movie shows a confocal image series of AC9-EGFP overexpressed in a HEK293 cell. Cell was treated with 10 μM isoproterenol added at t = 0 in the time stamp. AC9 recruitment to internal puncta is observed over the course of 30 min.
https://elifesciences.org/articles/58039#video1

each microscopic field that contained at least 10 such punctae. Both metrics verified selective internalization of the β2AR but not AC1 (*Figure 1C and D*, left set of bars).

We applied a similar tagging strategy to AC9 and verified that Flag tagging also does not disrupt the functional activity of AC9 (*Figure 1—figure supplement 1A,B*). Flag-AC9 localized predominantly in the plasma membrane in the absence of agonist, similar to Flag-AC1. However, after application of isoproterenol, Flag-AC9 redistributed to intracellular punctae, the majority of which also contain internalized HA-β2AR (*Figure 1B*). Punctate redistribution of both Flag-AC9 and HA-β2AR was also evident in lower-magnification widefield images (*Figure 1—figure supplement 1E*) and was verified quantitatively (*Figure 1C and D*, right set of bars). AC9 labeled in its C-terminal cytoplasmic domain with GFP (AC9-GFP) also redistributed, enabling live-cell confocal imaging which revealed mobile AC9-containing endosomes (*Video 1*). These results indicate that AC9 traffics dynamically to endosomes containing β2ARs, this trafficking is isoform-specific because AC1 remains in the plasma membrane, and it is regulated because AC9 accumulation in endosomes is increased by β2AR activation.

Isoproterenol also stimulated selective internalization of Flag-AC9 in the absence of recombinant β2AR overexpression (*Figure 1E and F*, *Figure 1—figure supplement 1C*) and this effect was blocked by the β-adrenergic antagonist alprenolol (*Figure 1G and H*, *Figure 1—figure supplement 1C*). These results indicate that endogenous β2AR activation is sufficient to stimulate AC9 trafficking and this is not an off-target drug effect. AC9, and ACs in general, are naturally expressed in low abundance. We were unable to reliably detect endogenous AC9 in HEK293 cells using available antibodies, but endogenous AC9 was detectable in primary human airway smooth muscle cells that naturally coexpress β2ARs (*Billington et al., 1999*). Endogenous AC9 immunoreactivity localized in these cells primarily to the plasma membrane under basal conditions, and its localization to internal punctae increased after isoproterenol application (*Figure 1I*). These results suggest that the trafficking behavior revealed by study of recombinant, tagged AC9 is relevant to the native protein.

## AC9 traffics via a similar pathway as β2AR but is differentially regulated

Internalized β2ARs accumulate in endosomes marked by Early Endosome Antigen 1 (EEA1) and agonist-dependent activation of Gs by the β2AR occurs on this compartment (*Irannejad et al., 2013*). We verified isoform-specific localization of Flag-AC9 but not Flag-AC1 to EEA1-marked endosomes by confocal microscopy (*Figure 2A and B*) and then applied anti-EEA1 immunoisolation (*Cottrell et al., 2009*; *Hammond et al., 2010*; *Temkin et al., 2011*) to purify these endosomes and probe their composition. Coexpressed HA-β2AR and Flag-AC9 were enriched in the endosome fraction prepared from isoproterenol-treated cells, as detected by immunoblot analysis and verified quantitatively across multiple experiments. In contrast, HA-β2AR but not Flag-AC1 was enriched in parallel isolations from cells co-expressing HA-β2AR and Flag-AC1, with similar levels of overall expression across both cell populations verified in cell lysates (*Figure 2C,D*, and *Figure 2—figure supplement 1B*). Documenting separation efficiency and fraction purity, the endosome fraction recovered ~34% of total cellular EEA1 but ≤5% of Golgi, endoplasmic reticulum, or plasma membrane markers (*Figure 2—figure supplement 1A*).

As a distinct and additional biochemical approach to verify these trafficking properties, we used cell surface biotinylation to assess protein depletion from the plasma membrane (*Flesch et al., 1995*; *Whistler and von Zastrow, 1998*). Isoproterenol produced a marked reduction of Flag-AC9 in the surface-biotinylated fraction but Flag-AC1 was unchanged, and surface HA-β2AR decreased after isoproterenol application irrespective of whether AC1 or AC9 was coexpressed (*Figure 2E,F*, and *Figure 2—figure supplement 1C*). These observations independently demonstrate isoform-specific trafficking of AC9 from the plasma membrane to endosomes, regulated coordinately with endocytosis of the β2AR.

A characteristic property of the conserved, clathrin-dependent pathway mediating β2AR endocytosis is that it also requires dynamin, an endocytic GTPase which can be acutely inhibited by the cell-permeant small molecule DYNGO-4a (*Irannejad et al., 2013*; *Macia et al., 2006*; *McCluskey et al., 2013*). Whereas isoproterenol promoted both Flag-β2AR and AC9-GFP to accumulate in endosomes in the vehicle (0.1% DMSO) control condition (*Figure 2G,H*, and *Figure 2—figure supplement 1D*), endosomal accumulation of both proteins was blocked in the presence of DYNGO-4a (*Figure 2I and J*, and *Figure 2—figure supplement 1E*). These results further support the hypothesis that

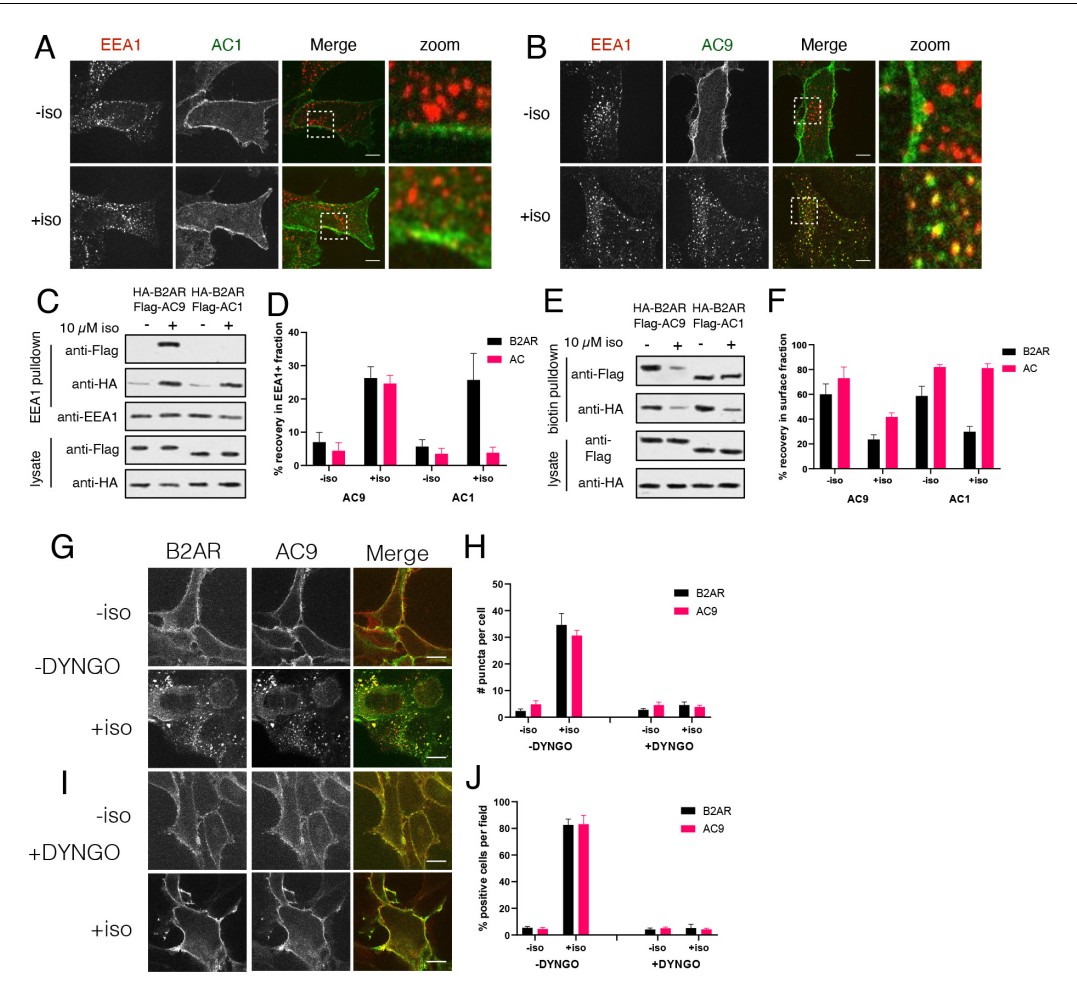

**Figure 2.** AC9 undergoes dynamin-dependent endocytosis and accumulates in endosomes marked by EEA1. (A–B) Representative confocal images of HEK293 cells expressing Flag-AC1 (A) or Flag-AC9 (B) after treatment with 10 µM isoproterenol or control for 30 min and stained for endogenous EEA1. Scale bar is 8 µm. (C) Representative western blot of a fraction isolated using antibodies to EEA1. Lanes 1–2 correspond to control HEK293 cells, lanes 3–4 to cells coexpressing Flag-AC9 and HA-β2AR, and lanes 5–6 to cells coexpressing Flag-AC1 and HA-β2AR. (D) Quantification of recovery of HA-β2AR, Flag-AC9 and Flag-AC1 in the endosome fraction relative to cell lysate. [mean ± SEM; n = 7 experiments]. **p<0.01 by two-tailed t-test. (E) Representative western blot of the surface exposed fraction isolated by surface labeling with Sulfo-NHS-biotin and purified with streptavidin. Lanes 1–2 correspond to cells coexpressing Flag-AC9 and HA-β2AR, and lanes 3–4 to cells coexpressing Flag-AC1 and HA-β2AR. (F) Quantification of recovery of HA-β2AR, Flag-AC9 and Flag-AC1 in the surface biotinylated fraction relative to total cell lysate. [mean ± SEM; n = 7 experiments]. **p<0.01 by two-tailed t-test. (G) Representative confocal images of HEK293 cells coexpressing HA-β2AR and Flag-AC9 after treatment with 10 µM isoproterenol or control for 30 min. Cells were treated with DMSO for 15 min prior to agonist exposure. (H) Quantification of internal puncta that are β2AR or AC9 positive, taken from wide field images (see *Figure 2—figure supplement 1D and E*) [mean ± SEM; n = 3 experiments, 10 visual fields and 200+ cells per condition]. **p<0.01 by two-tailed t-test. (I) Representative confocal images of HEK293 cells coexpressing HA-β2AR and Flag-AC9 after treatment with 10 µM isoproterenol or control for 30 min. Cells were treated with DYNGO-4a for 15 min prior to agonist exposure. (J) Quantification of cells with >10 internal puncta that are β2AR or AC1/9 positive, taken from wide field images (see *Figure 2—figure supplement 1D and E*) [mean ± SEM; n = 3 experiments, 10 visual fields and 200+ cells per condition]. **p<0.01 by two-tailed t-test.

The online version of this article includes the following figure supplement(s) for figure 2:

**Figure supplement 1.** Assessment of endosome fraction purity and wide field images of DYNGO-4a effect.

regulated AC9 trafficking to endosomes utilizes a shared membrane pathway and mechanism relative to regulated endocytosis of the β2AR, and many other GPCRs.

Despite these similarities, AC9 and β2AR were found to traffic independently. An early clue to this distinction was that brief exposure of cells outside of the tissue culture incubator (see Materials and methods) inhibits AC9 trafficking but β2AR trafficking is resistant to this environmental stress.

While we still do not fully understand the basis for this difference, it focused our attention on investigating the mechanism of AC9 traffic control in detail.

## AC9 trafficking is stimulated by GPCRs coupled to Gs but not Gi

We first asked if the β2AR is unique in its ability to stimulate endosomal accumulation of AC9 or if AC9 trafficking can be stimulated by another Gs-coupled GPCR. To do so we focused on the vasopressin-2 receptor (V2R), a Gs-coupled GPCR which also undergoes agonist-induced trafficking to early endosomes but is not endogenously expressed in HEK293 cells (*Birnbaumer, 2000*). The V2R is of particular interest because this GPCR, unlike the β2AR, has been shown to produce a sustained cAMP response via Gs that is associated with slow recycling and binding to β-arrestin in endosomes (*Innamorati et al., 1998*, *Feinstein et al., 2013*; *Klein et al., 2001*; *Oakley et al., 1999*; *Thomsen et al., 2016*). When examined in cells not transfected with recombinant V2Rs, Flag-AC9 remained in the plasma membrane irrespective of the presence of the V2R agonist arginine vasopressin (AVP) as expected (*Figure 3—figure supplement 1A*). However, in cells coexpressing HA-V2R, AVP stimulated the redistribution of both Flag-AC9 and HA-V2R to a shared population of endosomes (*Figure 3A,D,E*, *Figure 3—figure supplement 1D*). AVP-stimulated endocytosis of both proteins was confirmed by surface biotinylation (*Figure 3F,G*, *Figure 3—figure supplement 1B*). These results indicate that the ability of GPCR activation to promote trafficking of AC9 to endosomes is not unique to the β2AR. Rather, it appears to be a shared property of GPCRs which activate Gs.

We next asked if the ability to stimulate AC9 trafficking extends to GPCRs that couple to other heterotrimeric G proteins. We focused on the μ-opioid receptor (MOP-R or MOR) because this GPCR transits a similar early endocytic pathway as the β2AR (*Keith et al., 1998*) but activates Gi rather than Gs (*Kieffer and Evans, 2009*), and because opioid receptors have been shown explicitly to undergo ligand-dependent activation in endosomes (*Stoeber et al., 2018*). Application of the μ-opioid agonist [D-Ala$^2$, N-MePhe$^4$, Gly-ol]-enkephalin (DAMGO) stimulated transfected HA-MOR to accumulate in endosomes, as shown previously, while Flag-AC9 remained in the plasma membrane (*Figure 3B,D,E*, *Figure 3—figure supplement 1E*). We verified selective internalization of HA-MOR, but not Flag-AC9, by surface biotinylation (*Figure 3F,G*, *Figure 3—figure supplement 1C*). These results suggest that the ability to stimulate AC9 trafficking to endosomes is a property specific to GPCRs which couple to Gs relative to Gi.

We then returned to the V2R, coexpressing a mutant version truncated in its C-terminal cytoplasmic tail. This mutant V2R (HA-V2R-T) retains the ability to activate Gs at the plasma membrane but internalizes less efficiently after agonist-induced activation, promotes β-arrestin recruitment to endosomes less strongly and recycles more rapidly (*Innamorati et al., 1998*; *Innamorati et al., 1997*; *Oakley et al., 1999*). Despite visibly reduced internalization of HA-V2R-T after agonist application (*Figure 3C,D,E*), Flag-AC9 internalization was still observed (*Figure 3C,D,E* and *Figure 3—figure supplement 1F*) and this was confirmed by surface biotinylation (*Figure 3F,G*). These results suggest that Gs-coupled GPCRs share the ability to stimulate AC9 trafficking, irrespective of differences in receptor trafficking kinetics or binding to β-arrestin in endosomes.

## AC9 trafficking is not dependent on cytoplasmic cAMP or adenylyl cyclase activity

A possible basis for such shared control of AC9 trafficking is through cAMP elevation that occurs downstream of Gs activation. To test this, we applied the diterpene drug forskolin (FSK) to stimulate cytoplasmic cAMP production independently from the receptor or Gs. While AC9 is relatively insensitive to activation by FSK, other AC isoforms that are major contributors to cAMP production in HEK293 cells (such as AC3 and AC6) are sensitive, making FSK an effective stimulus of overall cAMP elevation (*Baldwin et al., 2019*; *Soto-Velasquez et al., 2018*). FSK did not cause detectable internalization of either HA-β2AR or Flag-AC9 assessed by imaging (*Figure 4—figure supplement 1A,C, D,I*) or surface biotinylation (*Figure 4—figure supplement 1E,F*). Further, as expected, Flag-AC1 remained in the plasma membrane irrespective of the presence of FSK (*Figure 4—figure supplement 1B–F,J*). This was also true in the combined presence of 3-isobutyl-1-methylxanthine (IBMX), a phosphodiesterase inhibitor which enhances FSK-induced cAMP elevation in the cytoplasm (*Figure 4—figure supplement 1H*).

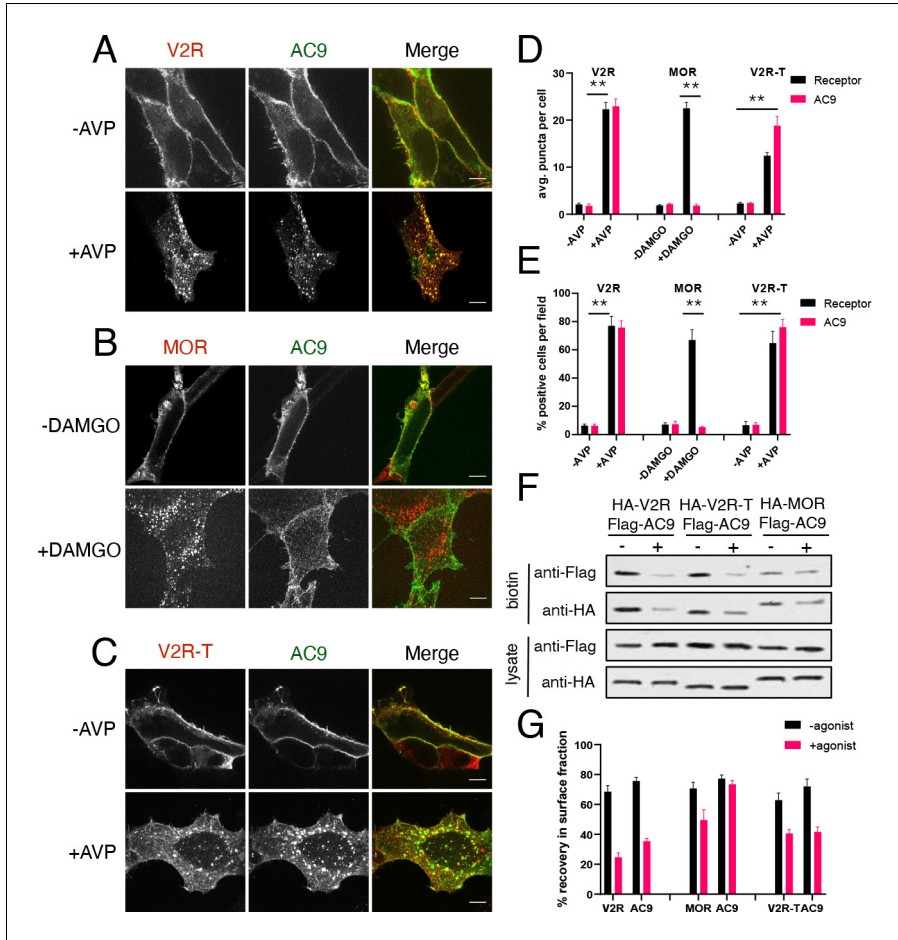

**Figure 3.** Gs but not Gi -coupled GPCRs promote AC9 trafficking. (A–C) Representative confocal imaging of HEK293 cells coexpressing Flag-AC9 and HA-V2R (A), HA-MOR (B), or HA-V2R-trunc (C), after treatment with 10 µM agonist (AVP or DAMGO) or control for 30 min. Scale Bar is 8 µm. (D) Quantification of internal puncta that are V2R, MOR, V2R-T, or AC1/9 positive, taken from wide field images (see *Figure 3—figure supplement 1D, E and F*) [mean ± SEM; n = 3 experiments, 10 visual fields and 200+ cells per condition]. **p<0.01 by two-tailed t-test. (E) Quantification of cells with >10 internal puncta that are V2R, MOR, V2R-T or AC1/9 positive, taken from wide field images (see *Figure 3—figure supplement 1D, E and F*) [mean ± SEM; n = 3 experiments, 10 visual fields and 200 + cells per condition]. **p<0.01 by two-tailed t-test. (F) Representative western blot of the surface biotinylated fraction from HEK293 cells coexpressing HA-V2R and Flag-AC9 (lanes 1–2), HA-V2R-T and Flag-AC9 (lanes 3–4), or HA-MOR and Flag-AC9 (lanes 5–6). (G) Recovery of tagged protein in the surface biotinylated fraction relative to the total cell lysate as seen in (F) [mean ± SEM; n = 7 experiments]. **p<0.01 by two-tailed t-test.

The online version of this article includes the following figure supplement(s) for figure 3:

**Figure supplement 1.** Validation that V2R and V2R-T activation stimulate AC9 trafficking.

As an independent approach, and to consider the possibility that cAMP exerts local rather than global control, we asked if endosomal accumulation of AC9 requires its own catalytic activity. To test this, we mutated a conserved aspartic acid residue that coordinates a catalytic magnesium in the active site, and which is essential for activity of AC6 (*Gao et al., 2011*; *Tesmer, 1997*). We verified that mutating the equivalent residue in AC9 (Flag-AC9-D442A) blocked cAMP production (*Figure 1—figure supplement 1B*), but found that regulated trafficking of Flag-AC9-D442A still occurred (*Figure 4—figure supplement 1G*). Together, these results indicate that the ability of GPCR-Gs activation to regulate AC9 trafficking is not a consequence of global cytoplasmic cAMP elevation, nor does it require local cAMP production by AC9.

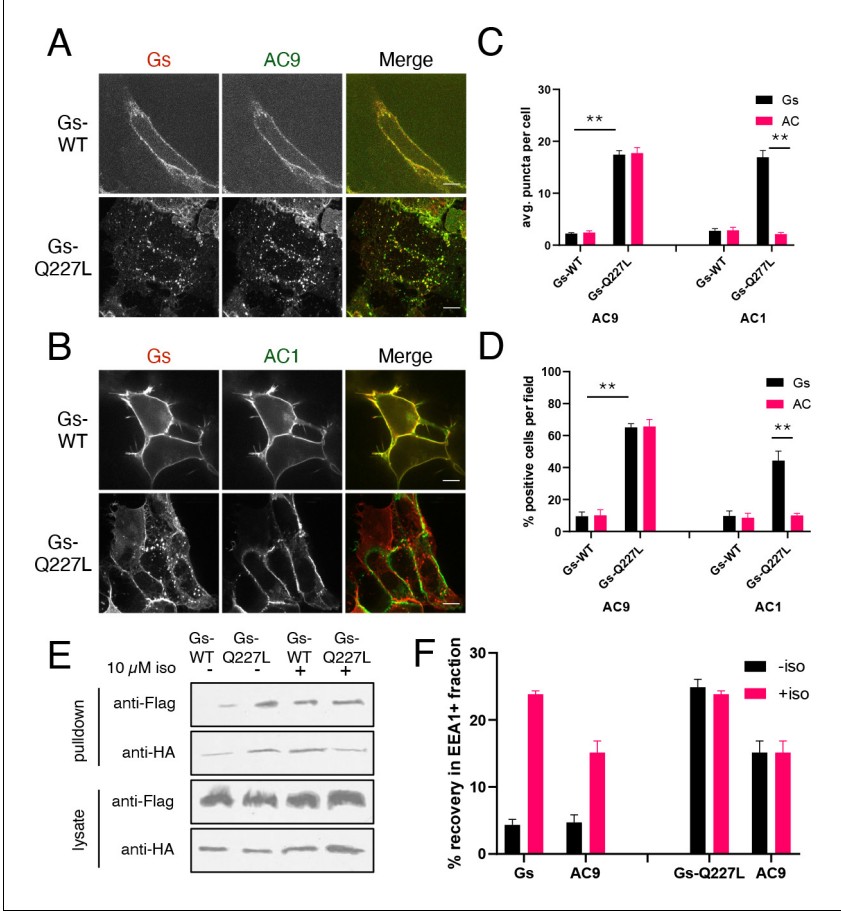

**Figure 4.** Gs activation is sufficient to promote AC9 internalization. (**A**) Representative confocal imaging of HEK293 cells cells coexpressing Flag-AC9, $G_s\beta$, $G_s\gamma$ and either HA-$G_s\alpha$ (HA-Gs) or HA-$G_s\alpha$-CA (HA-GsCA). (**B**) Representative confocal imaging of HEK293 cells coexpressing Flag-AC1, $G_s\beta$, $G_s\gamma$ and either HA-$G_s\alpha$ (HA-Gs) or HA-$G_s\alpha$-CA (HA-GsCA). (**C**) Quantification of internal puncta that are Gs or AC1/9 positive, taken from wide field images (see *Figure 4—figure supplement 2C and D*) [mean ± SEM; n = 3 experiments, 10 visual fields and 200+ cells per condition]. **p<0.01 by two-tailed t-test. (**D**) Quantification of cells with >10 internal puncta that are Gs or AC1/9 positive, taken from wide field images (see *Figure 4—figure supplement 2C and D*) [mean ± SEM; n = 3 experiments, 10 visual fields and 200+ cells per condition]. **p<0.01 by two-tailed t-test. (**E**) Representative western blot of an EEA1+ fraction from HEK293 cells coexpressing Flag-AC9 and HA-Gs (lanes 1 and 3) or Flag-AC9 and HA-GsCA (lanes 2 and 4) and after treatment with 10 µM isoproterenol (lanes 1–2) or control (lanes 3–4) for 30 min. (**F**) Quantification of the fraction of Flag-AC9 and HA-Gs/HA-GsCA recovered in the EEA1+ fraction (**E**) relative to total cell lysate. [mean ± SEM; n = 3 experiments] *p<0.05 **p<0.01 by two-tailed t-test.

The online version of this article includes the following figure supplement(s) for figure 4:

**Figure supplement 1.** Confirmation that forskolin does not promote AC9 trafficking.

**Figure supplement 2.** Gs accumulation in endosomes and verification that mutational activation of Gs is sufficient to stimulate AC9 trafficking.

**Figure supplement 3.** Effect of cholera toxin on AC9 trafficking.

## Gs activation is sufficient to promote AC9 trafficking

We next investigated whether AC9 internalization is regulated by Gs itself, and did so by introducing a point mutation into the alpha subunit (HA-Gs-Q227L) that renders Gs constitutively active by reducing its rate of intrinsic GTP hydrolysis (*Masters et al., 1989*). Flag-AC9 localized to the plasma membrane in the absence of agonist when coexpressed with wild type HA-Gs, but coexpression with activated HA-Gs-Q227L resulted in both proteins localizing to internal punctae (*Figure 4A,C,D*, *Figure 4—figure supplement 2E*). This effect was specific to AC9 because AC1 remained at the plasma membrane when coexpressed with either HA-Gs or HA-Gs-Q227L (*Figure 4B,C,D*,

*Figure 4—figure supplement 2F*). We verified by immunoisolation that both HA-Gs-Q227L and Flag-AC9 accumulate in EEA1-positive endosomes when coexpressed (*Figure 4—figure supplement 2A,B*), whereas coexpression of HA-Gs-Q227L with Flag-AC1 failed to produce endosome enrichment of either protein (*Figure 4—figure supplement 2C,D*). Independently supporting a discrete regulatory effect of Gs, application of cholera toxin (CTX) to activate the endogenous cellular complement of Gs resulted in receptor-independent accumulation of Flag-AC9, but not Flag-AC1, in endosomes (*Figure 4—figure supplement 3*).

The immunoisolation analysis also indicated that constitutive activation of Gs produced a degree of endosomal enrichment of AC9 similar to that produced in response to endogenous β2AR activation with isoproterenol. Further, application of isoproterenol to cells which coexpress HA-Gs-Q227L did not detectably increase the degree of endosomal enrichment observed for either Gs or AC9 (*Figure 4E,F*). Together, these results suggest that Gs activation mediated by endogenous β2ARs is fully sufficient to stimulate AC9 trafficking to endosomes, without requiring additional effects of upstream receptor activation or downstream cAMP signaling.

## Regulation of AC9 trafficking requires Gs but not β-arrestin

Because Gs activation is sufficient to stimulate accumulation of AC9 in endosomes, we next asked if it is necessary to regulate this trafficking process. To do so, we utilized previously described Gs-knockout (GsKO) cells which lack Gs due to CRISPR-mediated editing of the alpha subunit (GNAS) gene (*Stallaert et al., 2017*). Flag-β2AR and AC9-EGFP localized to the plasma membrane of GsKO as well as wild type HEK293 cells. However, AC9-GFP internalization was blocked in GsKO cells while Flag-β2AR still internalized. Moreover, AC9 trafficking was rescued by expression of recombinant HA-Gs (*Figure 5A-D*, *Figure 5—figure supplement 1A,B*). These results indicate that Gs is necessary for regulated trafficking of AC9 but not β2AR.

Stimulation of β2AR endocytosis by agonists is known to depend on β-arrestins (*Ferguson et al., 1996*; *Goodman et al., 1996*). Accordingly, we asked if this is also true for AC9. To test this, we used gene-edited HEK293 cells lacking both β-arrestin isoforms (Arrestins 2 and 3, or β-arrestin-1 and β-arrestin-2) (*O'Hayre et al., 2017*). Isoproterenol-stimulated internalization of HA-β2AR was lost in β-arrestin double-knockout (Arr DKO) cells, as expected, but AC9-EGFP internalization was still observed. Further, expressing recombinant Arrestin 3 (β-arrestin-2) rescued the HA-β2AR trafficking defect without a noticeable change in AC9-EGFP trafficking (*Figure 5E-H*, *Figure 5—figure supplement 1C,D*). These results indicate that AC9 and GPCR trafficking are regulated coordinately but through distinct mechanisms– with AC9 requiring Gs but not β-arrestin, and the GPCR requiring β-arrestin but not Gs.

## Functional evidence for AC9 signaling from endosomes

Both AC1 and AC9 are known physiological effectors of β-adrenergic signaling (*Sadana and Dessauer, 2009*; *Small et al., 2003*; *Tantisira et al., 2005*) and both are endogenously expressed in HEK293 cells, despite neither being the primary contributor to global cAMP elevation produced by β2AR activation in this cell type. Nevertheless, analysis by isoform-specific knockdown using siRNA indicated that both AC1 and AC9 make a small but statistically significant contribution to the overall cAMP elevation elicited by activation of endogenous β2ARs. AC1 but not AC9 depletion reduced the FSK-induced cAMP response (*Figure 6—figure supplement 1*), consistent with AC9 being relatively insensitive to stimulation by FSK (*Baldwin et al., 2019*) and verifying specificity of the knockdown approach. Considering that AC9 selectively accumulates in endosomes relative to AC1, we next investigated the hypothesis that AC9 also selectively contributes to the endosome-initiated component of the β2AR-elicited cellular cAMP response.

We tested this hypothesis using a pharmacological approach based on the ability of the membrane-impermeant β2AR antagonist CGP12177 (CGP) to access receptors selectively at the plasma membrane, whereas the membrane-permeant antagonist alprenolol accesses receptors both at the plasma membrane and endosomes (*Staehelin et al., 1983*). This approach has been used successfully in previous studies to isolate effects of endosomal activation (*Irannejad et al., 2013*; *Thomsen et al., 2016*). We validated it in the present study using a conformational biosensor, Nb80-EGFP, which is recruited specifically and reversibly by activated β2ARs in living cells (*Irannejad et al., 2013*). Isoproterenol application promoted recruitment of Nb80-GFP both to the

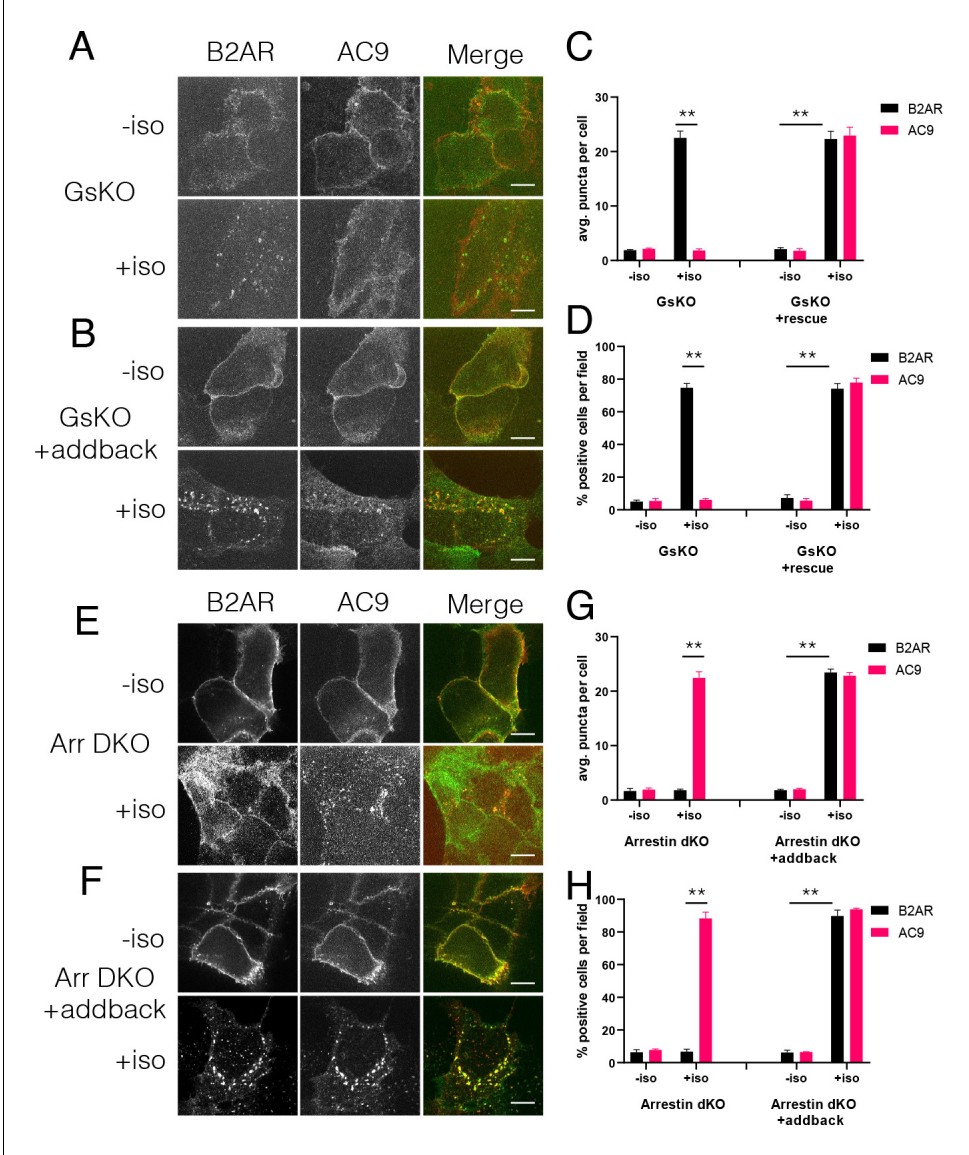

**Figure 5.** Regulation of AC9 internalization requires Gs but not beta-arrestin. (A–B) Representative confocal imaging of Gs-knockout (GsKO) HEK293 cells coexpressing Flag-β2AR, AC9-EGFP, and either pcDNA3 (A) or wild-type HA-Gs rescue (B). (C–D) Representative confocal imaging of Arrestin2/3 (β-arrestin −1 and 2) double knockout (Arr DKO) cells coexpressing Flag-β2AR, AC9-EGFP, and either pcDNA3 (C) or HA-Arrestin three rescue (D). (E) Quantification of internal puncta that are β2AR or AC9 positive, taken from wide field images (see *Figure 5—figure supplement 1A and B*) [mean ± SEM; n = 3 experiments, 10 visual fields and 200+ cells per condition]. **p<0.01 by two-tailed t-test. (F) Quantification of cells with >10 internal puncta that are β2AR or AC9 positive, taken from wide field images (see *Figure 5—figure supplement 1A and B*) [mean ± SEM; n = 3 experiments, 10 visual fields and 200+ cells per condition]. **p<0.01 by two-tailed t-test. (G) Quantification of internal puncta that are β2AR or AC9 positive, taken from wide field images (see *Figure 5—figure supplement 1C and D*) [mean ± SEM; n = 3 experiments, 10 visual fields and 200+ cells per condition]. **p<0.01 by two-tailed t-test. (H) Quantification of cells with >10 internal puncta that are Gs or AC1/9 positive, taken from wide field images (see *Figure 5—figure supplement 1C and D*) [mean ± SEM; n = 3 experiments, 10 visual fields and 200+ cells per condition]. **p<0.01 by two-tailed t-test.

The online version of this article includes the following figure supplement(s) for figure 5:

**Figure supplement 1.** Wide field images supporting distinct trafficking effects of Gs KO and Arr DKO.

plasma membrane and endosomes, and application of excess alprenolol rapidly reversed this activation readout at both locations (*Figure 6A*, *Video 2*). Application of CGP, in contrast, reversed Nb80-GFP recruitment only at the plasma membrane but not endosomes (*Figure 6B*, *Video 3*).

We next applied this approach to probe the contribution of endosomal β2AR activation to the overall cellular cytoplasmic cAMP response. Both CGP and alprenolol markedly inhibited the isoproterenol-induced cAMP elevation measured at 37 °C in living cells. This verifies that a large fraction of the overall cAMP elevation elicited by endogenous β2AR activation in these cells is initiated from the plasma membrane. However, we also consistently observed a more pronounced inhibition of cellular cAMP elevation following application of alprenolol compared to CGP (*Figure 6C*, left set of bars, and *Video 4*). We interpret this difference as a readout of the component of cAMP production initiated from endosomes. Remarkably, this CGP-resistant 'signal gap' was lost after AC9 knockdown but it remained in cells depleted of AC1 (*Figure 6C,D*, *Figure 6—figure supplement 1C–H*).

As another test of this hypothesis, and to investigate selectivity under conditions of recombinant AC overexpression (which were necessary for the trafficking studies), we asked if similar selectivity can be observed also using tagged AC isoforms. To do so we utilized gene-edited HEK293 cells lacking both AC3 and AC6 (AC3/6 DKO), which were shown previously to provide a reduced background useful for assessing effects of recombinant AC expression on cellular cAMP (*Soto-Velasquez et al., 2018*). The increment of isoproterenol-induced cAMP accumulation produced by overexpressing Flag-AC9 in these cells ('AC9 cAMP response') was blocked by alprenolol but not CGP (*Figure 6E*). In contrast, the corresponding increment produced by overexpressing Flag-AC1 ('AC1 cAMP response') was blocked by both alprenolol and CGP (*Figure 6F*). This verifies that AC9 selectively contributes to cellular cAMP production initiated by β2AR activation in endosomes using recombinant, as well as endogenous AC9.

## Discussion

The endocytic network is a dynamically regulated system critical for homeostatic integrity of the cell. From the point of view of GPCR-G protein signaling, this network was believed for many years to be silent, functioning only in signal termination and longer-term modulation of surface receptor number. Such homeostatic effects indeed occur, but an accumulating body of evidence supports an expanded view in which internalized GPCRs reacquire the ability to activate G proteins after endocytosis and initiate a second wave of signaling from endomembrane sites (*Irannejad et al., 2015*; *Lohse and Calebiro, 2013*; *Vilardaga et al., 2014*). Endosomal signaling depends on the presence of a G protein-regulated effector, but whether or how effectors localize to relevant internal membrane locations has remained a relatively unexplored frontier.

We approached this frontier by focusing on ACs as important effectors of signaling initiated by GPCR - Gs activation. We demonstrate dynamic and regulated trafficking of AC9 to early endosomes. This compartment is known to accumulate a wide variety of GPCRs, and it has been explicitly shown to be a site of Gs activation by the β2AR (*Irannejad et al., 2013*). AC9 is widely expressed (*Premont et al., 1996*), is a physiologically and clinically relevant effector of β2AR-Gs signaling in particular (*Small et al., 2003*; *Sunahara et al., 1996*; *Tantisira et al., 2005*) and is endogenously expressed in the HEK293 model system used in the present study. We also show that AC9, while contributing only a minor fraction to the overall cellular cAMP response elicited by β2AR activation in this system, is necessary to produce a specific endosome-initiated component of the endogenous β2AR-elicited cAMP response. Further, we demonstrate that AC9 is sufficient to increase cAMP production from endosomes when expressed as a recombinant protein. Moreover, we show that AC trafficking is isoform-specific because AC1 does not detectably accumulate in endosomes, nor does AC1 contribute detectably to the endosome-initiated component of cellular cAMP signaling (*Figure 6G*).

An important future goal is to identify structural and biochemical determinants of isoform-specific AC trafficking. We note that various isoform-specific protein interactions which impact other aspects of AC organization and function are already known, with AC9 being a particularly well-studied example (*Baldwin et al., 2019*). Another important question for future investigation is whether regulated intracellular trafficking is unique to AC9 or more widespread. We favor the latter possibility because a distantly related AC isoform was previously localized to a multivesicular intracellular compartment in *D. discoideum* (*Kriebel et al., 2008*). However, in this case, AC trafficking appears to occur

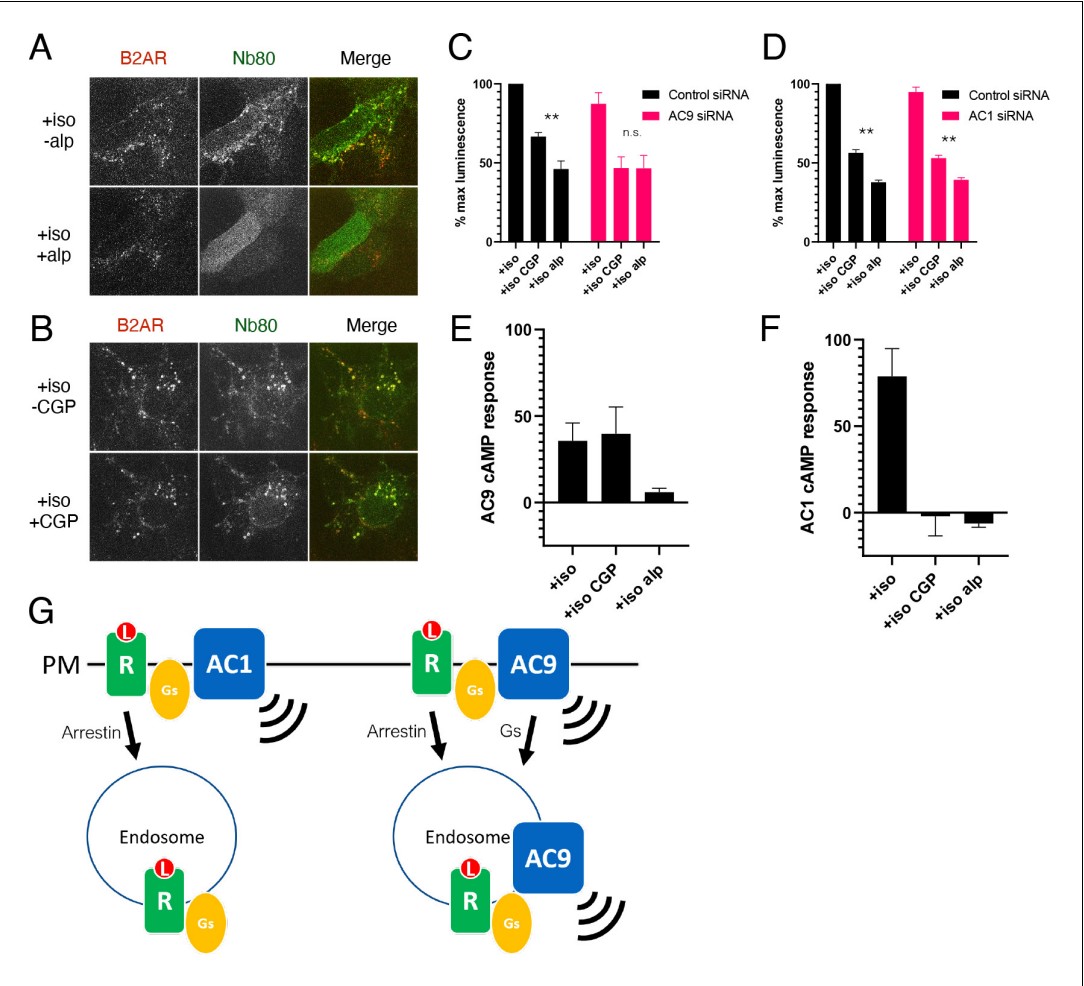

**Figure 6.** AC9 selectively contributes to the β2AR-mediated cAMP response from endosomes. (**A**) Recruitment of conformational biosensors to β2AR-containing endosomes is reversed by application of the membrane permeable antagonist alprenolol for 20 min. Scale Bar is 8 μm. See **Video 2** for full image series. (**B**) Recruitment of conformational biosensors to β2AR-containing endosomes is unaffected by application of the membrane impermeable antagonist CGP12177 for 20 min. Scale Bar is 8 μm. See **Video 3** for full image series. (**C**) Quantification of the maximum cAMP response in control and in AC9 siRNA knockdown HEK293 cells pretreated with 100 nM isoproterenol and exposed to supersaturating conditions of membrane permeable antagonist (10 μM alprenolol) or membrane impermeable antagonist (10 μM CGP12177). [mean ± SEM; n = 4 experiments] (**D**) Quantification of the maximum cAMP response in control in AC1 siRNA knockdown HEK293 cells pretreated with 100 nM isoproterenol and exposed to supersaturating conditions of membrane permeable antagonist (10 μM alprenolol) or membrane impermeable antagonist (10 μM CGP12177). [mean ± SEM; n = 4 experiments] (**E**) Quantification of the maximum cAMP response in AC3/6KO HEK293 cells due to AC9 overexpression as the delta between the Flag-AC9 and pcDNA3 conditions. Cells were pretreated with 100 nM isoproterenol and exposed to supersaturating conditions of membrane permeable antagonist (10 μM alprenolol) or membrane impermeable antagonist (10 μM CGP12177). [mean ± SEM; n = 4 experiments] (**F**) Quantification of the maximum cAMP response in AC3/6KO HEK293 cells due to AC1 overexpression as the delta between the Flag-AC1 and pcDNA3 conditions. Cells were pretreated with 100 nM isoproterenol and exposed to supersaturating conditions of membrane permeable antagonist (10 μM alprenolol) or membrane impermeable antagonist (10 μM CGP12177). [mean ± SEM; n = 4 experiments] (**G**) Model: Ligand binding causes initial signaling event at the PM followed by arrestin-dependent endocytosis of β2AR. AC1 is restricted to the PM but AC9 is dynamically redistributed by a distinct Gs regulated process, and contributes to the β2AR-mediated cAMP response from the endosome. The online version of this article includes the following figure supplement(s) for figure 6:

**Figure supplement 1.** Verification that AC9 selectively contributes to cAMP production stimulated by endogenous receptor activation in endosomes.

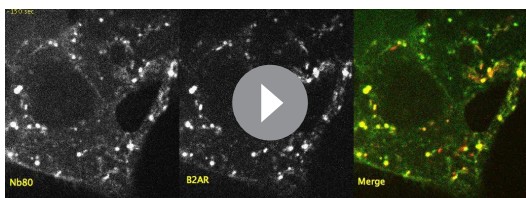

**Video 2.** This movie shows a confocal image series of β2AR (red) and Nb80-EGFP (green) from a HEK293 cell pre-incubated for 20 min with 100 nM isoproterenol. 10 µM alprenolol, added at t = 0 time point indicated in the time stamp, reverses Nb80-EGFP recruitment to β2ARs at both the plasma membrane and endosomes.
https://elifesciences.org/articles/58039#video2

through the biosynthetic pathway and it is not known if the AC-containing compartment also contains a relevant GPCR or G protein. We also note that several other transmembrane AC isoforms have been implicated previously in endomembrane cAMP signaling by mammalian GPCRs (*Calebiro et al., 2009*; *Cancino et al., 2014*; *Ferrandon et al., 2009*; *Kotowski et al., 2011*; *Mullershausen et al., 2009*; *Vilardaga et al., 2014*), and that a distinct AC isoform which lacks any transmembrane domains ('soluble' AC or AC10) has been implicated as well (*Inda et al., 2016*). Thus we anticipate that AC9 is not the only isoform to exhibit discrete trafficking behavior, and that much remains to be learned along this line. In particular, we note that the localization and trafficking properties of AC3 and AC6– which are major contributors to overall cAMP production stimulated by β2ARs in HEK293 cells (*Soto-Velasquez et al., 2018*)– have yet to be delineated.

One possible mechanism of AC9 trafficking to GPCR-containing endosomes is by physical association with the receptor or receptor-G protein complex, and there is previous evidence indicating that AC5 can form a complex including GPCRs (*Navarro et al., 2018*). However, our results provide two lines of evidence indicating that AC9 traffics independently, despite trafficking via a similar dynamin-dependent membrane pathway as the β2AR and in a coordinated manner. First, activation of Gs is sufficient to promote the accumulation of AC9 but not β2AR in endosomes. Second, AC9 trafficking requires Gs but not β-arrestins, whereas the converse is true for trafficking of the β2AR. Accordingly, AC trafficking is likely subject to different modulatory input(s) relative to the trafficking of GPCRs. This is consistent with the difference in environmental sensitivity between AC9 and β2AR trafficking which initially motivated our investigations. However, additional studies will be required to fully elucidate the mechanistic basis for differential control of AC9 trafficking, and to delineate physiological inputs into regulated AC trafficking more broadly. The physiological significance of isoform-specific AC trafficking also remains to be determined, but we note that there is already significant evidence that cAMP produced internally can mediate different downstream signaling effects relative to cAMP produced from the plasma membrane (*O'Banion et al., 2019*; *Tsvetanova and von Zastrow, 2014*).

In closing, to our knowledge the present study is the first to delineate the dynamic endocytic trafficking of a functionally relevant AC isoform, and to identify a role of Gs in regulating

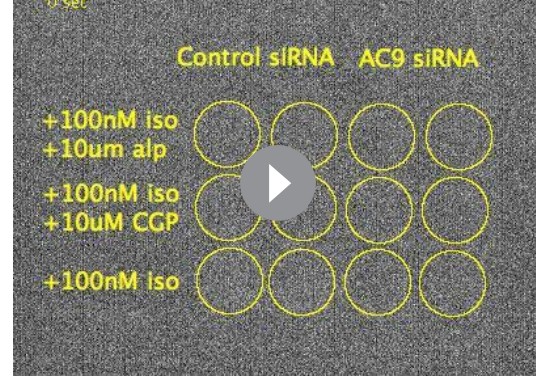

**Video 4.** This movie shows an image series of the luminescence-based cAMP biosensor from HEK293 cells that have been treated with control siRNA (columns 1 and 2) or AC9 siRNA (columns 3 and 4). Cells were preincubated with 100 nM isoproterenol for 20 min prior to imaging, and 10 µM CGP12177 or 10 µM alprenolol were added immediately before imaging, where indicated.
https://elifesciences.org/articles/58039#video4

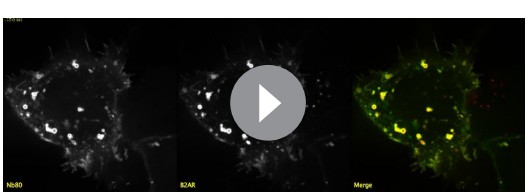

**Video 3.** This movie shows a confocal image series of β2AR (red) and Nb80-EGFP (green) from a HEK293 cell pre-incubated for 20 min with 100 nM isoproterenol. 10 µM CGP12177, added at t = 0 indicated in the time stamp, reverses Nb80-EGFP recruitment to β2ARs at the plasma membrane but not at endosomes.
https://elifesciences.org/articles/58039#video3

the trafficking of a defined AC separately from its catalytic activity. The finding that such AC trafficking is isoform-specific, and regulated separately from its activating GPCR, reveals a new layer of specificity and control in the cAMP system.

# Materials and methods

## Key resources table

| Reagent type (species) or resource | Designation | Source or reference | Identifiers | Additional information |
|---|---|---|---|---|
| Cell line (*Homo sapiens*) | HEK293 | ATCC | CRL-1573; RRID:CVCL_0045 | Human embryonic kidney (female) |
| Cell line (*Homo sapiens*) | GNAS-knockout | *Stallaert et al., 2017* | | HEK293 parental |
| Cell line (*Homo sapiens*) | Arrestin 2 and 3 double knockout | *O'Hayre et al., 2017* | | HEK293 parental |
| Cell line (*Homo sapiens*) | ADCY 3 and 6 double knockout | *Soto-Velasquez et al., 2018* | | HEK293 parental |
| Biological sample (*Homo sapiens*) | Human smooth airway muscle cells (HSAMs) | Prepared from lung biopsy (*Tsvetanova et al., 2017*) | | Primary cell culture |
| Antibody | Mouse anti-FLAG (M1) | Sigma-Aldrich | F-3040; RRID:AB_439712 | (1:1000) |
| Antibody | Rabbit anti-Flag | Sigma-Aldrich | F7425; RRID:AB_439687 | (1:1000) |
| Antibody | Mouse anti-HA | Biolegend | 16B12; RRID:AB_2820200 | (1:1000) |
| Antibody | Goat anti-AC9 | Santa Cruz Biotech | sc-8576; RRID:AB_2223286 | (1:50) |
| Antibody | Mouse anti-EEA1 | Fisher Scientific | 610457; RRID:AB_397830 | (1:1000) |
| Antibody | Mouse anti-Golgin-97 | Thermo | A-21270; RRID:AB_221447 | (1:1000) |
| Antibody | Rabbit anti-calnexin | Cell Signaling | 2679P; RRID:AB_2228381 | (1:1000) |
| Antibody | Mouse anti-Sodium/ Potassium ATPase | Novus Biologicals | NB300-540SS; RRID:AB_306023 | (1:1000) |
| Recombinant DNA reagent (human) | Flag-AC9 (plasmid) | *Paterson et al., 2000* | | pcDNA3 backbone |
| Recombinant DNA reagent (human) | Flag-AC1 | *Chen et al., 1997* | | pcDNA3 backbone |
| Recombinant DNA reagent (human) | Flag-AC9-D442A | This study | | pcDNA3 backbone |
| Recombinant DNA reagent (human) | AC9-EGFP | This study | | EGFP-C1 backbone |
| Recombinant DNA reagent (human) | HA-B2AR | *von Zastrow and Kobilka, 1992* | | pcDNA3 backbone |
| Recombinant DNA reagent (human) | HA-V2R | *Rochdi et al., 2010* | | pcDNA3 backbone |
| Recombinant DNA reagent (mouse) | HA-MOR | *Whistler and von Zastrow, 1998* | | pcDNA3 backbone |
| Recombinant DNA reagent (human) | HA-V2R-T | *Rochdi et al., 2010* | | pcDNA3 backbone |
| Recombinant DNA reagent | Nb80-EGFP | *Irannejad et al., 2013* | | EGFP-C1 backbone |
| Recombinant DNA reagent (human) | HA-G(alpha)s | *Irannejad et al., 2013* | | pcDNA3 backbone |

*Continued on next page*

*Continued*

| Reagent type (species) or resource | Designation | Source or reference | Identifiers | Additional information |
|---|---|---|---|---|
| Recombinant DNA reagent (human) | G(beta-1) | *Irannejad et al., 2013* | | pcDNA3 backbone |
| Recombinant DNA reagent (human) | G(gamma-2) | *Irannejad et al., 2013* | | pcDNA3 backbone |
| Recombinant DNA reagent (human) | HA-G(alpha)s -Q227L | *Masters et al., 1989* | | pcDNA3 backbone |
| Recombinant DNA reagent | pGloSensor-20F | Promega | E1171 | |
| Transfected construct (human) | AC9-siRNA | This study | | See Materials and methods for sequence |
| Transfected construct (human) | AC1 siRNA | This study | | See Materials and methods for sequence |
| Chemical compound, drug | Isoprenaline (iso) | Sigma-Aldrich | 51-30-9 | |
| Chemical compound, drug | Arginine-vasopressin (AVP) | Sigma-Aldrich | 113-79-1 | |
| Chemical compound, drug | DAMGO, [D-Ala2, N-Me-Phe4, Gly5-ol]-Enkephalin acetate salt | Sigma-Aldrich | E7384 | |
| Chemical compound, drug | DYNGO-4a | Abcam Biochemicals | ab120689 | |
| Chemical compound, drug | Alprenolol | Sigma | 3707-88-5 | |
| Chemical compound, drug | CGP12177 | Tocris | 1134 | |
| Chemical compound, drug | Forskolin | Sigma-Aldrich | 66575-29-9 | |
| Chemical compound, drug | 3-isobutyl-1-methylxanthine (IBMX) | Sigma-Aldrich | 28822-58-4 | |
| Chemical compound | D-Luciferin, sodium salt | Gold Biosciences | LUCNA-1G | |
| Chemical compound | Anti-mouse IgG Magnetic microbeads | Miltenyi | 130-047-101 | |
| Chemical compound | EZ-link Sulfo-NHS-biotin | Pierce | 21425 | |
| Commercial assay or kit | Alexa Fluor 647 Protein Labeling Kit | Thermo Fisher Scientific | A20173 | |
| Commercial assay or kit | Alexa Fluor 488 Protein Labeling Kit | Thermo Fisher Scientific | A20181 | |
| Commercial assay or kit | Direct cAMP ELISA | Enzo Life Sciences | ADI-900–066 | |
| Software, algorithm | Prism | GraphPad | 8.1.1 | |
| Software, algorithm | ImageJ | Imagej.net | 2.0.0-rc-69/1.52 p | |
| Software, algorithm | MATLAB | MathWorks | R2014b | |

## Cell culture, expression constructs, and transfections

HEK 293 cells (CRL-1573, ATCC, mycoplasma-tested) were cultured in complete growth Dulbecco's modified Eagle's medium (DMEM, Gibco) and supplemented with 10% fetal bovine serum (UCSF Cell Culture Facility). HA-β2AR (*Tang et al., 1999*; *von Zastrow and Kobilka, 1992*), HA-V2R (*Rochdi et al., 2010*), HA-MOR (*Whistler and von Zastrow, 1998*), HA-V2R-T (*Charest and Bouvier, 2003*; *Rochdi et al., 2010*), all described previously, were sub-cloned from Flag-tagged

constructs. Nb80-EGFP was previously described (*Irannejad et al., 2013*). HA-G(alpha)s, G(beta-1), G(gamma-2) were gifts from Philip Wedegaertner. HA-G(alpha)s-Q227L, a previously described point mutant of Gs that is constitutively active (*Masters et al., 1989*), was made from the original construct using the QuikChange Site-Directed Mutagenesis Kit (Agilent Technologies) with the forward primer 5'-CGATGTGGGCGGCCTGCGCGATGAACGCCGC-3'. Flag-AC1, Flag-AC9 from the Dessauer Lab, were originally described by *Hacker et al., 1998*; *Krupinski et al., 1989*; *Paterson et al., 2000*; *Premont et al., 1996*. Flag-AC9-D442A (Catalytic inactive mutant) was also made from the original construct using QuikChange Kit with the forward primer 5'-CCACTAG TCCAGTGTGGTGGAATTCGCCATGGACTACAAAGACGATGACGAC-3'. Transfections were carried out using Lipofectamine 2000 (Life Technologies) according to the manufacturer's protocol. Cells were transfected 48 hr before experiments. siRNA knockdown of AC1 and AC9 expression in HEK293 cells was carried out using Lipofectamine RNAiMAX (Life Technologies) according to the manufacturer's protocol. Cells were transfected 72 hr before experiments. Knockdown of AC1 used the siRNA CCGGGCGGTTCAGACCTTCAA and AC9 knockdown used CTGGGCATGAGGAGG TTTAAA.

Primary cultures of human airway smooth muscle cells were prepared as described previously (*Tsvetanova et al., 2017*). Cells were passaged no more than five times using Trypsin-EDTA (Life Technologies) and maintained in 10% FBS in DMEM.

Gs knockout (*Stallaert et al., 2017*) and beta-arrestin-1/2 double knockout (*O'Hayre et al., 2017*) HEK293 cells were previously described. AC3/AC6 double knockout HEK293 cells were also described previously (*Soto-Velasquez et al., 2018*) and were provided as a generous gift by Drs. Monica Soto-Valasquez and Val Watts (Purdue University). Cells were passaged using PBS-EDTA and maintained in 10% FBS in DMEM.

Cholera Toxin (Sigma) was administered to cells for 16 hr overnight treatment at 10 ng/ml concentration in 10% FBS in DMEM.

We found AC9 trafficking to be environmentally sensitive. Specifically, exposure of cells outside of the incubator for more than 2 min tended to reduce the degree of isoproterenol-stimulated internalization of AC9, without affecting internalization of β2AR. Accordingly, this restriction was consistently adhered to in the present study.

## Antibodies

Antibodies used were rabbit anti-Flag (Sigma), mouse anti-Flag M1 (Sigma), mouse anti-Flag M2 (Sigma), mouse anti-HA 16B12 (Biolegend), rat anti-HA (Roche), goat anti-AC9 (Santa Cruz Biotech), mouse anti-Golgin-97 (Thermo), rabbit anti-calnexin (Cell Signaling), mouse anti-Sodium/Potassium ATPase (Fisher).

## Fixed cell confocal imaging

Cells were transfected with the indicated construct(s) and then plated on glass coverslips coated with poly-L-lysine (0.0001%, Sigma) 24 hr later. For antibody feeding assays, cells were: (1) placed on ice and rinsed with ice-cold phosphate-buffered saline (PBS), (2) labeled by the addition of antibodies diluted 1:1000 in DMEM for 10 min, and (3) rinsed with room temperature PBS and allowed to traffic for 30 min by the addition of 37°C fresh media (DMEM + 10% fetal bovine serum) with or without a saturating concentration of β2AR agonist (10 μM isoproterenol, Sigma), V2R agonist (10 μM arginine-vasopressin, Sigma), MOR agonist (10 μM DAMGO, Sigma), or forskolin (10 μM, Sigma). For all assays, cells were rinsed with cold PBS and fixed by incubation in 3.7% formaldehyde (Fisher Scientific) diluted in modified BRB80 buffer (80 mM PIPES, 1 mM $MgCl_2$, 1 mM $CaCl_2$, pH 6.8) for 20 min at room temperature. Cells were then blocked in 2% Bovine Serum Albumin (Sigma) in PBS with permeabilization by 0.2% triton X-100 (Sigma). Primary antibody labeling was performed by the addition of antibodies diluted 1:1000 in blocking/permeabilization buffer for one hour at room temperature. Secondary labeling was performed by addition of the following antibodies diluted at 1:500 in blocking/permeabilization buffer for 20 min at room temperature: Alexa Fluor 555 or 488 donkey anti-mouse (Invitrogen), Alexa Fluor 555 or 488 donkey anti-rabbit (Invitrogen), Alexa Fluor 488 or 555 goat anti-rat (Invitrogen), or Alexa Fluor 488 donkey anti-sheep (Life Technologies). Specimens were mounted using ProLong Gold antifade reagent (Life Technologies).

Fixed cells were imaged by spinning disc confocal microscope (Nikon TE-2000 with Yokogawa confocal scanner unit CSU22) using a 100X NA 1.45 objective. A 488 nm argon laser and a 568 nm argon/krypton laser (Melles Griot) were used as light sources.

## Microscope image acquisition and image analysis

Spinning disc images were collected using an electron multiplying CCD camera (Andor iXon 897) operated in the linear range controlled by Micro-Manager software (https://www.micro-manager.org). Images were processed at full bit depth for all analysis and rendered for display by converting to RGB format using ImageJ software (http://imagej.nih.gov/ij) and linear look up table. The number of endosomes was quantified by thresholding images and the ParticleTracker ImageJ plugin.

## Live-cell confocal imaging

Live cell imaging was carried out using Yokagawa CSU22 spinning disk confocal microscope with a × 100, 1.4 numerical aperture, oil objective and a $CO_2$°C and 37°C temperature-controlled incubator. A 488 nm argon laser and a 568 nm argon/krypton laser (Melles Griot) were used as light sources for imaging EGFP and Flag signals, respectively. Cells expressing both Flag-tagged receptor and the indicated nanobody–EGFP were plated onto glass coverslips. Receptors were surface labelled by addition of M1 anti-Flag antibody (1:1000, Sigma) conjugated to Alexa 555 (A10470, Invitrogen) to the media for 30 min, as described previously. Indicated agonist (isoprenaline, Sigma) or antagonist (CGP-12177, Tocris) (alprenolol, Sigma) were added and cells were imaged every 3 s for 20 min in DMEM without phenol red supplemented with 30 mM HEPES, pH 7.4 (UCSF Cell Culture Facility). Time-lapse images were acquired with a Cascade II EM charge-coupled-device (CCD) camera (Photometrics) driven by Micro-Manager 1.4 (http://www.micro-manager.org).

## Endosome immunoisolation

Cells were transfected with the indicated construct(s) 48 hr before lysis and plated onto 60 mm cell culture dishes 24 hr before lysis. Cells were allowed to traffic for 30 min by the addition of 37°C fresh media (DMEM + 10% fetal bovine serum) with or without a saturating concentration of the indicated agonist. Cells were then placed on ice, washed with ice-cold PBS, and scraped into an isotonic homogenization buffer (10 mM HEPES, 100 mM KCl, 25 mM sucrose, Complete protease inhibitor (Roche), pH 7.2) and passaged 20 times through a 22 G BD PrecisionGlide Needle. Whole cell lysates were then spun down at 1000 G for 10 min at 4°C and the pellets discarded. The supernatant was then bound to Early Endosome Antigen one mouse antibody (1:250, Fisher Scientific) and anti-mouse IgG magnetic microbeads (Miltenyi Biotech) overnight. Endosomes were then bound to magnetic columns which were blocked with 3% BSA and washed with PBS. Proteins in the isolated fraction were eluted with 0.1% Triton-X and characterized by western blot.

## Surface biotinylation

Cells were transfected with the indicated construct(s) 48 hr before lysis and plated onto 60 mm cell culture dishes coated with poly-L-lysine (0.0001%, Sigma) 24 hr before lysis. Cells were allowed to traffic for 30 min by the addition of 37°C fresh media (DMEM + 10% fetal bovine serum) with or without a saturating concentration of the indicated agonist. Cells were then placed on ice, washed with ice-cold PBS, and then surface labeled with EZ-link Sulfo-NHS-biotin (Pierce) for 30 min, rocking at 4°C. Reaction was then quenched with tris buffered saline (TBS) twice for 10 min. Cells were then placed on ice, washed with ice-cold PBS, and scraped into an isotonic homogenization buffer (10 mM HEPES, 100 mM KCl, 25 mM sucrose, Complete protease inhibitor (Roche), pH 7.2) and passaged 20 times through a 22 G BD PrecisionGlide Needle. Cell lysate was then bound to streptavidin agarose resin (Thermo) overnight. Resin was spun down and the supernatant discarded, resuspended and washed in ice-cold PBS, and characterized by western blot.

## Real-time cAMP assay in living cells

Real-time analysis of cAMP elevations were carried out in living HEK293 cells and in the absence of phosphodiesterase inhibitors using a were transfected with a plasmid encoding a cyclic-permuted luciferase reporter construct, based on a mutated RIIβB cAMP-binding domain from PKA (pGloSensor-20F, Promega), which produces rapid and reversible cAMP-dependent activation of luciferase

activity in intact cells and is capable of detecting cAMP elevations in the absence of phosphodiesterase inhibitors. Cells were plated in 24-well dishes containing approximately 200,000 cells per well in 500 µl DMEM without phenol red and no serum and equilibrated to 37°C in a light-proof cabinet. An image of the plate was focused on a 512 × 512 pixel electron multiplying CCD sensor (Hamamatsu C9100-13), cells were equilibrated for 1 hr in the presence of 250 µg ml$^{-1}$ luciferin (Gold biosciences), and sequential luminescence images were collected every 10 s to obtain basal luminescence values. The camera shutter was closed, the cabinet opened and the indicated concentration of isoprenaline was bath applied, with gentle manual rocking before replacing in the dark cabinet and resuming luminescence image acquisition. In endocytic manipulation experiments, cells were pre-incubated with 30 µM Dyngo-4a (abcam Biochemicals) for 15 min. Every 10 s, sequential images were acquired using Micro-Manager (http://www.micro-manager.org) and integrated luminescence intensity detected from each well was calculated after background subtraction and correction for vignetting using scripts written in MATLAB (MathWorks). In each multiwell plate, and for each experimental condition, a reference value of luminescence was measured in the presence of 5 µM forskolin, a manipulation that stimulates a moderate amount of receptor-independent activation of adenylyl cyclase. The average luminescence value—measured across duplicate wells—was normalized to the maximum luminescence value measured in the presence of 5 µM forskolin.

## Biochemical assay of cAMP accumulation

A biochemical assay of cAMP accumulation was used to determine the effects of AC mutation on catalytic activity, with high sensitivity and without dependence on subcellular location due to inhibition of cellular phosphodiesterases. Briefly, cells were pre-incubated in the presence of 1 mM IBMX (Sigma) for 30 min at 37°C in Dulbecco's modified Eagle's medium followed, and then incubated for an additional 10 min in absence or presence of isoproterenol (in the continued presence of IBMX), as indicated. Cells were quickly washed with ice-cold PBS and lysed by exposure to 0.1 M HCl for 10 min at room temperature. The cAMP concentration in lysates was determined using a commercial immunoassay (Direct cAMP ELISA kit, Enzo Life Sciences, Farmingdale, NY) according to the manufacturer's instructions.

## Statistical analysis

Results are displayed as the mean of results from each experiment or data set, as indicated in figure legends. The statistical significance between conditions for experiments with two conditions was calculated using paired, two tailed t-tests. All statistical calculations were performed using Excel (Microsoft Office) or Prism (GraphPad). The threshold for significance was $p < 0.05$ and the coding for significance is reported as follows: (n.s.) $p > 0.05$, (*) $p \leq 0.05$, (**) $p \leq 0.01$.

# Acknowledgements

We thank B Lobingier, G Peng, L Ripoll, M Stoeber and other current and past members of the von Zastrow laboratory for advice and critical discussions. We thank R D Mullins, R Sunahara and J Taunton for valuable discussion and suggestions. We thank M Soto-Velasquez and V Watts for valuable discussion and the generous gift of AC3/6 knockout cells. We thank the E Roth group for their generous support and lab equipment, and K Kurtz for assistance. This work was supported by grants from the US National Institutes of Health (DA010711 and DA012864 to MvZ; GM60419 to CWD; HL124049 to AAS; CA209891 to JSG). AL is an ARCS scholar.

# Additional information

## Funding

| Funder | Grant reference number | Author |
| --- | --- | --- |
| National Institutes of Health | DA010154 | Mark Von Zastrow |
| National Institutes of Health | DA012864 | Mark Von Zastrow |
| National Institutes of Health | GM60419 | Carmen W Dessauer |
| National Institutes of Health | HL124049 | Aparna B Sundaram |

| National Institutes of Health | CA209891 | J Silvio Gutkind |
|---|---|---|
| National Institutes of Health | HL122508 | Roshanak Irannejad |
| National Institutes of Health | GM133521 | Roshanak Irannejad |

The funders had no role in study design, data collection and interpretation, or the decision to submit the work for publication.

## Author contributions

André M Lazar, Conceptualization, Data curation, Formal analysis, Validation, Investigation, Visualization, Methodology, Writing - original draft, Writing - review and editing; Roshanak Irannejad, Supervision, Validation, Investigation, Writing - review and editing; Tanya A Baldwin, Investigation, Methodology, Writing - review and editing; Aparna B Sundaram, Resources, Methodology; J Silvio Gutkind, Resources, Writing - review and editing; Asuka Inoue, Resources, Methodology, Writing - review and editing; Carmen W Dessauer, Conceptualization, Resources, Methodology, Writing - review and editing; Mark Von Zastrow, Conceptualization, Formal analysis, Supervision, Funding acquisition, Writing - original draft, Project administration, Writing - review and editing

## Author ORCIDs

André M Lazar  https://orcid.org/0000-0003-3372-3742
Roshanak Irannejad  https://orcid.org/0000-0001-8702-2285
Aparna B Sundaram  http://orcid.org/0000-0002-4076-4756
Mark Von Zastrow  https://orcid.org/0000-0003-1375-6926

## Decision letter and Author response

Decision letter https://doi.org/10.7554/eLife.58039.sa1
Author response https://doi.org/10.7554/eLife.58039.sa2

## Additional files

### Supplementary files

• Transparent reporting form

### Data availability

All data generated or analysed during this study are included in the manuscript and supporting files. Source data files have been provided all main figures.

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
