## [Decision Letter]

**Acceptance summary:**

This work carefully and convincingly demonstrates that Gs activation during β-adrenergic receptor signaling drives endocytosis of a specific isoform of a downstream effector, adenyl cyclase 9, which contributes to cAMP production.

**Decision letter after peer review:**

Thank you for submitting your article "G protein-regulated endocytic trafficking of adenylyl cyclase type 9" for consideration by *eLife*. Your article has been reviewed by three peer reviewers, and the evaluation has been overseen by a Reviewing Editor and Suzanne Pfeffer as the Senior Editor. The reviewers have opted to remain anonymous.

The reviewers have discussed the reviews with one another and the Reviewing Editor has drafted this decision to help you prepare a revised submission.

Summary:

This study addresses the question of adenyl cyclase (AC) endocytosis during GPCR signaling. Morphological and biochemical experiments primarily in transfected HEK293 cells show that two Gs-coupled GPCRs drive isoform specific endocytosis of AC9 and that Gs is required and sufficient while β-arrestin is not. The internalized AC9 contributes to cAMP production stemming from intracellular receptors. While Gs-induced AC9 endocytosis is independent of cAMP production, the actual mechanism linking Gs activation to AC9 endocytosis remains unknown.

Overall the work shows Gs activation during β-adrenergic receptor signaling drives endosomal localization of the downstream effector AC9. The reviews were positive. The experiments are well-controlled, show mostly clear effects, and are convincing for the system being investigated. Nevertheless, certain conceptual and technical issues were raised that need to be addressed prior to publication.

Essential revisions:

1) The β2AR is not the prototypical GPCR for studying sustained cAMP generation from endosomes. Although the authors present some data for the V2R that has been previously shown to induce very sustained cAMP responses, it is well-known that several class A or B GPCRs possess this feature as well and to a far greater extent than that observed for the β2AR. If at all possible, the study would benefit from control experiments using another GPCR class A (e.g., TSHR) or class B (e.g., GLP-1R, PAC1R, CTR), which would more broadly determine how generalizable these findings are. Additionally, the authors should put forth a rationale as to why they selected the β2AR for this study as opposed to the more canonical receptors mentioned above.

2) All fluorescence images shown in the main figures are confocal images, however the quantification data presented in the main figures were obtained from widefield images shown in supplementary figures. While confocal images show a clear internalization of both HA-β2AR and Flag-AC9, the widefield images have high levels of background, which can make particle tracking more difficult. Some concerns about this:

– What's the reason behind choosing to quantify images acquired by widefield and show representative confocal images? The type of microscopy used could affect particle tracking, especially when we consider the lower resolution in widefield compared to confocal.

– Widefield is a great choice for live cell imaging, however, most of these experiments were done in fixed cells. Confocal will show "better" endosomes. Quantification should be done with confocal images.

– While authors provided a satisfactory statistics analysis, all "representative" images and curves (for cAMP production) do not have an 'n'. How many times were these experiments repeated?

– Quantification of fluorescence images and representative images should be done with images acquired by the same microscopy method to avoid differences based on the nature of microscopy kind.

---

## [Author Response]

Essential revisions:1) The β2AR is not the prototypical GPCR for studying sustained cAMP generation from endosomes. Although the authors present some data for the V2R that has been previously shown to induce very sustained cAMP responses, it is well-known that several class A or B GPCRs possess this feature as well and to a far greater extent than that observed for the β2AR. If at all possible, the study would benefit from control experiments using another GPCR class A (e.g., TSHR) or class B (e.g., GLP-1R, PAC1R, CTR), which would more broadly determine how generalizable these findings are. Additionally, the authors should put forth a rationale as to why they selected the β2AR for this study as opposed to the more canonical receptors mentioned above.

We agree that the B2AR is not a prototype for sustained signaling. Instead, β-2 receptors provide an example of GPCRs which cycle rapidly and continuously through endosomes, activating Gs transiently each round. The B2AR is also widely recognized not to recruit β-arrestins to endosomes, in contrast to some other GPCRs with longer residence in endosomes which have been shown to mediate a sustained cAMP response.

However, we do not believe that these considerations negate the importance of endosomal signaling stimulated by the B2AR. Previously published data (Irannejad et al., 2013, cited in the text), showed that endosomal activation of the B2AR mediates a detectable (albeit moderate) contribution to (transient) global cAMP elevation. We have also shown that this is required for the cAMP-dependent transcriptional response elicited by endogenous B2AR activation (Tsvetanova et al., 2014, also cited). We independently verify in the present study that β-2 receptor activation in endosomes makes a significant (yet rapidly reversible) contribution to the global cellular cAMP response, and provide new data showing that this component of transient cAMP signaling requires AC9.

Nevertheless, we agree that the results summarized above do not necessarily mean that AC9 is also relevant to signaling by GPCRs that mediate sustained Gs activation, and that we failed to address in the initial submission. In the revised manuscript we explicitly note and cite that signaling initiated from endosomes can be transient or sustained (Introduction, second paragraph), and that the V2 vasopressin receptor (V2R) mediates sustained signaling (–subsection “AC9 trafficking is stimulated by GPCRs coupled to Gs but not Gi”, first paragraph). We demonstrate in the present study that V2Rs are indeed capable of driving AC9 trafficking in 293 cells when recombinantly expressed in these cells, similarly to recombinantly expressed B2ARs which define our primary focus.

We also show that a V2R truncation mutant can drive AC9 trafficking. We focused on this observation as an initial clue to a separation between receptor and AC9 trafficking. However this mutant receptor is also well known to lose its ‘class B’ phenotype (β-arrestin recruitment to endosomes) and recycle more rapidly than wild-type V2R. We have modified the text to explicitly point this out in the revised Results section (–subsection “AC9 trafficking is stimulated by GPCRs coupled to Gs but not Gi”, last paragraph). We believe that these results provide additional, independent support for our conclusion that AC9 trafficking is not restricted to the β-2 adrenergic receptor or to rapidly recycling / transiently signaling GPCRs. This is also consistent with the ability of receptor-independent activation of Gs (mutationally or using cholera toxin) to stimulate AC9 trafficking, as was already shown and discussed in the manuscript.

2) All fluorescence images shown in the main figures are confocal images, however the quantification data presented in the main figures were obtained from widefield images shown in supplementary figures. While confocal images show a clear internalization of both HA-β2AR and Flag-AC9, the widefield images have high levels of background, which can make particle tracking more difficult. Some concerns about this:– What's the reason behind choosing to quantify images acquired by widefield and show representative confocal images? The type of microscopy used could affect particle tracking, especially when we consider the lower resolution in widefield compared to confocal.– Widefield is a great choice for live cell imaging, however, most of these experiments were done in fixed cells. Confocal will show "better" endosomes. Quantification should be done with confocal images.– While authors provided a satisfactory statistics analysis, all "representative" images and curves (for cAMP production) do not have an 'n'. How many times were these experiments repeated?– Quantification of fluorescence images and representative images should be done with images acquired by the same microscopy method to avoid differences based on the nature of microscopy kind.

The reason that we show wide field images is to verify the trafficking behavior over large cell ‘lawns’ and to indicate that the trafficking phenotype is robust and consistent across cells. We also think this is useful because, as the reviewers indicate, widefield is less sensitive for detecting internalization effects relative to confocal (which can emphasize internal structures depending on the plane of section). We think the fact that one can easily see the trafficking phenotype with low-mag widefield imaging is significant and may be relevant for those investigators who lack routine access to a confocal microscope or objectives with sufficiently high magnification / numerical aperture to resolve individual endosomes as the reviewers indicate. Actually we do show (in the main figures) multiple examples of such higher-resolution images, but this is not the point of including the lower resolution images of cell lawns in the supplement. We also agree that quantifying more confocal images could provide additional support, but again this was not the purpose of including widefield images. In fact we believe that the (two) biochemical approaches used (surface biotinylation and endosome isolation) more incisively establish internalization of AC9. To clarify this, in the revised manuscript we more fully explained our rationale for showing lower-resolution images in data supplement (subsection “Regulated and isoform-selective trafficking of AC9 to endosomes”, second paragraph).